# Using an In Vivo Mouse Model to Determine the Exclusion Criteria of Preexisting Anti-AAV9 Neutralizing Antibody Titer of Pompe Disease Patients in Clinical Trials

**DOI:** 10.3390/v16030400

**Published:** 2024-03-05

**Authors:** Hanqing Wang, Cengceng Zhang, Zheyue Dong, Xueyang Zhu, Xuchu Zheng, Ziyang Liu, Jianfang Zhou, Shuangqing Yu, Xiaobing Wu, Xiaoyan Dong

**Affiliations:** 1Genecradle Therapeutics Inc., Beijing 100176, China; wanghq@bj-genecradle.com (H.W.); zhangcce@bj-genecradle.com (C.Z.); zhuxy@bj-genecradle.com (X.Z.); zhengxuchu2023@163.com (X.Z.); liuziyang@bj-genecradle.com (Z.L.); zhoujf@bj-genecradle.com (J.Z.); dongxiaoyan@bj-genecradle.com (X.D.); 2Beijing FivePlus Gene Technology Co., Ltd., Beijing 102629, China; dongzy@fiveplusgene.com.cn

**Keywords:** gene therapy, adeno-associated virus, preexisting neutralizing antibody, Pompe disease

## Abstract

The efficacy of adeno-associated virus (AAV)-based gene therapy is dependent on effective viral transduction, which might be inhibited by preexisting immunity to AAV acquired from infection or maternal delivery. Anti-AAV neutralizing Abs (NAbs) titer is usually measured by in vitro assay and used for patient enroll; however, this assay could not evaluate NAbs’ impacts on AAV pharmacology and potential harm in vivo. Here, we infused a mouse anti-AAV9 monoclonal antibody into Balb/C mice 2 h before receiving 1.2 × 10^14^ or 3 × 10^13^ vg/kg of rAAV9-coGAA by tail vein, a drug for our ongoing clinical trials for Pompe disease. The pharmacokinetics, pharmacodynamics, and cellular responses combined with in vitro NAb assay validated the different impacts of preexisting NAbs at different levels in vivo. Sustained GAA expression in the heart, liver, diaphragm, and quadriceps were observed. The presence of high-level NAb, a titer about 1:1000, accelerated vector clearance in blood and completely blocked transduction. The AAV-specific T cell responses tended to increase when the titer of NAb exceeded 1:200. A low-level NAbs, near 1:100, had no effect on transduction in the heart and liver as well as cellular responses, but decreased transduction in muscles slightly. Therefore, we propose to preclude patients with NAb titers > 1:100 from rAAV9-coGAA clinical trials.

## 1. Introduction

Adeno-associated virus (AAV)-based gene therapy is a promising treatment for inherited genetic diseases. Due to exposure to wild-type AAV or acquisition from maternal immunity, a relatively large proportion of humans carry circulating antibodies against the AAV capsid. Anti-AAV Neutralizing Abs (NAbs) could block viral infection and reduce the efficiency of tissue transduction with AAV vectors, even at a low titer [1,2]. Hence, patients with preexisting anti-AAV antibodies were excluded in most gene therapy clinical trials. However, in hemophilia B gene therapy, successful AAV5 transduction was achieved in patients with low levels of preexisting NAbs [3], which indicated that anti-AAV NAb-positive patients below a certain level might also benefit from gene therapy. A quantitative assessment of the role of preexisting NAbs in the efficacy and safety of the drug is required for the successful clinical development of AAV gene therapy.

Most data on liver-directed AAV8 or AAV5 capsid-based vectors have been documented and the transduction in the liver assessed using report genes or secreted proteins such as FIX [1,4,5]. Little is known about the impact of preexisting NAbs on the AAV9 vector, which has a broad tropism and transduces many cell types, including cardiomyocytes, hepatocytes, muscles, and the central nervous system [6]. AAV9-SMN1 (named Zolgensma^®^) has been approved for intravenous injection for spinal muscular atrophy (SMA) patients; nevertheless, its efficacy and safety data from anti-AAV9 antibody-positive patients are absent.

Pompe disease (PD) is a lysosomal glycogen storage disease caused by acid α-glucosidase (GAA) deficiency, which is characterized by glycogen accumulation in the heart, muscles, and central nervous system. The disease poses a range of clinical symptoms and outcomes, such as infantile-onset PD (IOPD), manifesting with hypertrophic cardiomyopathy, hypotonia, and respiratory insufficiency and progressing to death at early life, or late-onset PD (LOPD), presenting with a progressive myopathy in childhood or later [7,8]. AAV9 vector-mediated gene therapy is expected to introduce the functional GAA gene and produce the GAA enzyme continuously for PD patients. We developed a recombinant vector rAAV9-coGAA for the treatment of Pompe disease, named GC301 in brief. A single intravenous injection of GC301 resulted in an elevation of GAA enzyme activity, reduction of glycogen accumulation, and improvement of pathological changes in Pompe model mice [9]. These results support further clinical evaluation of GC301.

Here, we infused a mouse anti-AAV9 monoclonal antibody (MoAb) in naïve mouse serum into Balb/C mice 2 h before receiving 1.2 × 10^14^ or 3 × 10^13^ vg/kg GC301 and explored the role of NAbs on biodistribution and transduction of GC301 in the liver, heart, and muscles. The pharmacokinetics, pharmacodynamics, and cellular responses in combination with in vitro NAb assay validated the effects of preexisting NAbs. These parameters from in vivo models are used for establishing a threshold of preexisting anti-AAV9 NAb level from which acceptable efficacy and safety can be achieved in these NAb-positive patients in future clinical trials.

## 2. Materials and Methods

### 2.1. Vectors

rAAV9-coGAA, named GC301 in brief, consists of a codon-optimized human GAA that was transcribed and driven by a constitutive promoter [9]. GC301 used in the study was produced by a scaled production method based on the Bac-to-AAV package system and manufactured as described under GMP condition in Beijing Fiveplus Gene Technology Company (Beijing, China) [10]. Genome titers (i.e., vector genome copies [vg/mL]) of AAV vectors were determined by real-time PCR and with plasmid DNA as standards, as reported previously [9]. The vectors were qualified and characterized the same as batches used for clinical trials.

For the in vitro neutralization assay, an AAV9 vector encoding Gaussia luciferase (rAAV9-EGFP-2A-Gluc) was used as a reporter.

### 2.2. Cell Culture

HEK293 cell line (ATCC CRL-1573) was maintained under 37 °C, 5% CO_2_ condition in DMEM supplemented with 10% FBS (D10) and used for the AAV neutralization assay.

### 2.3. Animal Studies

Animal protocols were approved by GeneCradle’s Ethical Committee and conducted by certified operators according to the guidelines and regulations on animal experiments. In all experiments, female Balb/C mice aged 4~6 weeks were used and randomly assigned to each group, with 5~6 animals in each group. Animals were passively immunized by tail vein injection with 0~42 μg/mL of AAV9 specific mouse monoclonal antibody (anti-AAV9 MoAb) diluted with naïve mouse serum in a volume of 100 μL, and mice injected with PBS were used as negative control (NC). The anti-AAV9 MoAb tested in our study was generated by us using hybridoma technique from empty AAV9 particle-immunized Balb/C mice. The MoAb is a specifically neutralized AAV9 serotype with an IC50 of 20 ng/mL in vitro. After 2 h, mice received 1.2 × 10^14^ (namely high dose, H) or 3 × 10^13^ vg/kg (namely low dose, L) of GC301 by the tail vein route, which is the dose for IOPD (ClinicalTrials.gov Identifier: NCT05793307) and LOPD in clinical trials, respectively. Blood samples were collected from the retro-orbital plexus using capillary tubes before and after vector administration. Mice were euthanized at 4 or 7 weeks post-GC301 administration, and tissues were collected and snap-frozen for additional studies. Mouse spleens were harvested at 4 weeks post-GC301 injection, and cells were isolated freshly for ELISPOT assay.

### 2.4. Anti-AAV9 Neutralizing Antibody Assay In Vitro

Mouse serum samples were heat-inactivated at 56 °C for 30 min. Neutralization was measured as the reduction in Gluc reporter gene expression from rAAV9-EGFP-2A-Gluc when inhibited by the NAbs present in the serum samples in a 96-well plate. Serum was 2x serial diluted with D10 from 1:20 for a total of 6 to 8 dilutions and mixed with 2 × 10^8^ vg rAAV9-EGFP-2A-Gluc in 50 μL of DMEM containing 0.1% BSA (Sigma, St. Louis, MO, USA). One hour after incubation at 37 °C, a trypsinized single-cell suspension of 20,000 HEK-293 cells in 100 μL of D10 was added. The plates were then incubated at 37 °C under 5% CO_2_ for 48 to 72 h. After incubation, 20 μL of supernatant was aspirated and 50 μL of coelenterazine native substrate (Nanolight, Tempe, AZ, USA) was added to each well while protected from light. The relative luciferase unit (RLU) was measured using a GLOMAX 96 microplate luminometer (Promega, Madison, WI, USA). For each AAV NAb assay, negative serum with 2 μg/mL and 0.2 μg/mL of anti-AAV9 MoAb was used as the positive reference to establish the in-house standard, whereas pooled serum samples from naive mouse sera were used as the negative controls.

The neutralizing antibody titer was defined as the reciprocal of the serum dilution causing a 50% reduction of RLU compared to the rAAV9-EGFP-2A-Glu control. The transduction inhibition titer was analyzed by Prism software 8.0 (GraphPad Software, San Diego, CA, USA), and the titer at which a serum inhibited 50% of the transduction (IC50) was calculated using a nonlinear regression.

### 2.5. Vector Genome Copy Number Detected by Droplet Digital (dd) PCR

DNA was extracted from whole-organ homogenization or EDTA-Na2 anti-coagulate blood using TIANamp genomic DNA Kit (Tiangen Biotech, Beijing, China). The GC301 genome copy numbers in DNA of tissues and blood were determined using a digital droplet PCR assay with primers and probes, the same as that used for GC301 quantification. The primers and probe were coGAA-specific (forward primer 5′-GGCACCTGGTCGTGGAACTC-3′, reverse primer 5′-CCAGCCTCTGATCGGCAAGG-3′, probe [FAM] 5′-TGGCCAGGCTCCACCGCCTT-3′ [TAMRA]) and were synthesized by Sangon Biotech, Shanghai, China. Each sample was tested in duplicates; assay results were reported as mean genome copy number per μg genome DNA.

### 2.6. GAA Enzyme Activity Assay

Tissues were homogenized using stainless-steel beads in RIPA lysate buffer with protease inhibitors (Solarbio, Beijing, China). Protein concentrations were determined by Bradford assay. GAA activity in the tissue homogenates was measured at pH 3.9 by conversion of the substrate 4-methylumbelliferyl (4-MU) D-glucoside to the fluorescent product umbelliferone [11]. Data were represented as nanomoles of substrate cleaved in 1 h per milligram of total protein (nmol/mg/h). The tissues collected from negative control (received 100 μL of PBS) and positive control (received 1.2 × 10^14^ vg/kg of GC301) mice were concluded as process quality control.

### 2.7. ELISPOT

Peptide libraries (18–20 mer overlapping by 10~12 amino acids [aa]) covering the sequence of the AAV9 capsid VP1 protein or human GAA was synthesized by ChinaPeptide Qyaobio (Shanghai, China) and Scilight-Peptide (Beijing, China), respectively. Mouse spleens were harvested 4 weeks after GC301 injection, and spleen cells were isolated using EZ-SepTM mouse lymphocyte separation medium. Cells with confirmed viability > 90% were tested for antigen-specific responses to peptide pools. A total of 3 × 10^5^ cells were stimulated by the peptide pools with each peptide at the final concentration of 2.5 μg/mL. PMA and Ionomycin were included as a positive control stimulation for all of the samples; mock stimulation (0.25% DMSO) was included as a negative control. All 5 conditions were plated in duplicate for each sample and stimulated for 30–36 h at 37 °C, 5% CO_2_. Mouse IFN-γ precoated ELISpot Kit (Dakewe Biotech, Shenzhen, China) were prepared and IFN-γ spot-forming units (SFUs) were developed according to the manufacturer’s instructions. SFUs in each well were enumerated using an automated spot counter (ImmunoSpot CTL S6 Micro Analyzer, Cellular Technology Limited, Shaker Heights, OH, USA). The final results were reported as SFU/million cells. An antigen-specific response equal to or greater than 50 SFUs/million cells and 2-fold over the mock response was positive.

### 2.8. Statistical Analysis

One-way or two-way ANOVA was performed to determine whether the means and variances were equal across datasets, and the statistical differences between groups were determined by multiple comparisons or two-tailed Student’s *t* test using GraphPad Prism software (version 8). Values of *p* < 0.05 were considered statistically significant. Data are expressed as mean ± SD or mean and 95% CI.

## 3. Results

### 3.1. Effective Transduction of Systemic GC301 Delivery in Multiple Tissues in Balb/C Mice

As the dose of 1.2 × 10^14^ vg/kg was selected for the ongoing clinical trial for IOPD, we originally injected Balb/C mice at 4~6 weeks with GC301 at this dose via tail vein. The mice injected with PBS were used as negative control (NC). The pharmacodynamic biomarker, GAA activity, was analyzed for the kinetics study and evaluation of transduction efficiency of GC301. We measured the major organs, heart, and liver as well as skeletal muscle, including diaphragm and quadriceps, at 4, 6, 8, and 12 weeks post-injection (wpi.). In NC group, the GAA activity kept at low level (Figure 1A). Around 20~40 nmol/mg/h was found in liver and below 20 nmol/mg/h in heart, diaphragm, and quadriceps throughout the tested time points. In GC301-treated group, significantly enhanced GAA activity was detected in the tissues at 4 wpi (Figure 1B). The peak of GAA expression in liver, diaphragm, and quadriceps occurred at Week 6 and then gradually declined. A sustained increase was found in heart and up to 6000 nmol/mg/h (6647 ± 1611 nmol/mg/h, *n* = 5) at 12 wpi. The liver GAA activity maintained at the level of 2000 nmol/mg/h at 12 wpi. and those of diaphragm and quadriceps were also 20–50-fold as compared with NC at 12 wpi (502 ± 168 vs. 8 ± 5 nmol/mg/h and 259 ± 162 vs. 10 ± 2 nmol/mg/h, respectively). Our data suggested that the systemic GC301 delivery could introduce the GAA gene into a series of important organs and tissues, which are also the majorly affected tissues in PD, where persisting GAA expressed and functioned. Therefore, high levels of GAA activity in the heart and liver could be used as the indexes for investigating the effects of preexisting NAbs.

### 3.2. In Vitro Neutralizing Assay Validating Ab Titers of a Mice Monoclonal NAb in Human or Mouse Sera Pool and That Delivered In Vivo

We established a cell-based in vitro AAV9-transduction inhibition assay using an anti-AAV9 MoAb as the in-house standard. The assay limit of detection (LOD) was 2.7 ng/mL for the anti-AAV9 MoAb and 1:20 for serum. To compare the matrix effect of human and mouse sera on in vitro NAb assay, we spiked the anti-AAV9 MoAb in pooled mouse or human negative sera at the concentrations used in our experiments, then measured their titers. The titers showed the minor difference between the titers in pooled negative sera from humans and mice (Table 1). Next, we delivered 100 μL of spiked mouse sera containing 0~42 μg/mL anti-AAV9 MoAb via tail vein; the sera samples were collected from mice 2 h later, and the titers were measured. The titer of the mice sera was 1:52~1:69, 1:158~1:255, 1:176~1:479, 1:377~1:612, and 1:658~1:1101 at 2 h after spiked sera infusion at the MoAb concentration of 1.5, 3, 7, 21, and 42 μg/mL, respectively. The mimicking antisera titers range from <1:100 to above 1:1000, which covered the range of preexisting AAV9 NAb in human.

### 3.3. Accelerated Vector Clearance in Peripheral Blood in Presence of High Level of Preexisting NAb

1.2 × 10^14^ vg/kg of GC301 was delivered 2 h after anti-AAV9 MoAb infusion by intravenous injection; the mice infused with pooled negative mice sera (0 μg/mL MoAb-transferring group) were included as control. We measured the vector genome copies in blood at 2, 7, 14, and 28 days after injection (Figure 2). The peak of GC301 genome copies in blood was found at day 2 which was the first time point and the vector genomes were 1.86 × 10^8^ ± 7.85 × 10^7^, 1.70 × 10^8^ ± 1.23 × 10^8^, 1.06 × 10^8^ ± 9.93 × 10^7^, and 3.89 × 10^5^ ± 3.78 × 10^5^ copies/μg DNA in the 0, 7, 21, and 42 μg/mL MoAb-pretreated groups, respectively. The kinetics of vector clearance show similar patterns in the tested groups, and gene copies decreased rapidly in the first week and gradually from Day 7 to Day 28. The significantly lowest level of vector copy number was found in the 42 μg/mL MoAb-pretreated group as compared with other groups throughout the assayed time points. The corresponding NAb titers in this group of immunized mice measured by our in vitro assay is around 1:1000.

### 3.4. Decreased Vector Genome Copies and Transduction of GC301 in Heart and Liver in Presence of Moderate Levels of Preexisting NAb

We further analyzed the vector genome copies and its transductions in heart and liver 4 weeks post-GC301 injection. As demonstrated in Figure 3A,B, the vector genome in the liver and heart of non-MoAb-transferring mice was 3.88 × 10^6^ ± 1.79 × 10^6^ and 1.25 × 10^5^ ± 3.75 × 10^4^ copies/μg DNA, respectively. The low levels of NAb at 1.5 and 3 μg/mL has no reduction in vector genome in either the liver or heart, whereas the moderate levels of NAb at 7 and 21 μg/mL groups reduced the vector genome in liver more than 50%, down to 1.79 × 10^6^ ± 1.06 × 10^6^ and 2.18 × 10^5^ ± 1.64 × 10^5^ copies/μg in the liver, respectively. The reduction in heart was unremarkable, remaining at 8.02 × 10^4^ ± 1.79 × 10^4^ copies/μg and 3.91 × 10^4^ ± 2.86 × 10^4^ copies/μg in the group of 7 μg/mL and the 21 μg/mL group, respectively.

Subsequently, we compared the transductions of GC301 in presence of low levels of preexisting NAb, 1.5 and 3 μg/mL. We discovered that the GAA activity in liver pretreated by 1.5 μg/mL and 3 μg/mL MoAb decreased to 53.13% and 38.39% of that of the non-MoAb-transferring group, respectively (*p* < 0.0001, Figure 3C). No inhibition on GAA activity in heart was observed at these groups (Figure 3D).

### 3.5. Elicited AAV9- and GAA-Specific T Cell Responses

Considering a potential effect of antigen–antibody complexes on immune responses, we assayed specific T cell responses by measuring the IFN-γ production from peptides-stimulated spleen cells at 4 weeks after receiving GC301. Most of the mice had no response to GAA peptides, and the spot count was <50 SFUs per million cells, except for 2/5 in the 3 μg/mL group and 1/5 in the 7 and 42 μg/mL MoAb-pretreated groups, which were positive, with SFUs slightly above 50 per million cells (Figure 4A). AAV9 capsid-specific T cell response was positive, and ≥50 SFUs per million cells were seen in most mice, especially in groups with the transfusion of MoAb ≥ 3 μg/mL (Figure 4B). Although T cell response showed no significant difference among groups treated with 3, 7, 21, and 42 μg/mL, the AAV-specific T cell responses tend to increase with the enhanced titers of preexisting NAb. Stronger T cell responses were induced by AAV capsid peptides than those by GAA peptides, implying a relatively poor immunogenicity of GAA introduced in target cells.

### 3.6. Efficient Transductions of GC301 in 1.5 μg/mL MoAb-Pretreated Mice at Both High and Low Dosages

As demonstrated above, the data was obtained from 1.2 × 10^14^ vg/kg of GC301, namely the high dose group (H), which almost is the highest dosage used in clinical studies till now. The GAA activity in the liver confirmed the inhibition of Nab in groups above 3 μg/mL, which might reduce the amount of GAA enzyme secreted into the blood and the absorption by peripheral organs. Therefore, we assessed the potential impact of a low level of preexisting Nab by injection with 100 μL of 1.5 μg/mL MoAb on lower vector dosage, 3 × 10^13^ vg/kg, namely the low dose group (L). The mice without preexisting antibody were included as a control. After GC301 administration, GAA activity in the heart, liver, diaphragm, and quadriceps were measured at 7 weeks post-injection. No significant effect of preexisting NAb on GAA enzyme activity in the heart (Figure 5A) and liver (Figure 5B) was observed at 7 weeks after administration in both the L and H groups. However, in the H group, the GAA activity decreased 38.3% (188.4 ± 58.8 nmol/mg/h vs. 305.3 ± 168.3 nmol/mg/h, *p* > 0.05) and 57.3% (145.9 ± 38.4 nmol/mg/h vs. 342.0 ± 167.1 nmol/mg/h, *p* < 0.01) in the diaphragm (Figure 5C) and quadriceps (Figure 5D), respectively, in MoAb-pretreated as compared to non-MoAb-pretreated groups. No differences of GAA activity in the diaphragm and quadriceps were observed between the L groups with or without preexisting NAb. Despite the fact that vector transduction was reduced slightly in muscles compared with antibody-negative animals at the high dose, significant GAA activity enhancements were observed after GC301 administration.

## 4. Discussion

The extensive clinical experiences and the approvals of Luxturna, Zolgensma, HEMGENIX, and Roctavian have attested AAV vectors to be the best options for in vivo gene delivery [12,13]. Despite these successes, not all patients are eligible candidates for this novel approach, and its application was recommended for the AAV immunity-naïve population. Natural AAV infection might occur at a very early stage of life and serum anti-AAV Abs in infants could be obtained from the mother. The estimated seroprevalence for NAb for the different AAV serotypes ranges from 30–90% in the population [14,15,16,17,18]. The presence of NAbs could block AAV transduction and enhance the immunogenicity of AAV vectors; thus, many efforts including capsid engineering, concomitant with immunosuppressive agents, plasma dialysis, or imlifidase treatment to remove immunoglobulin have been made to avoid or prevent immune responses [19,20,21]. In vitro assay measuring anti-AAV NAbs is usually an initial test for patient enrollment, though it cannot evaluate the impacts of NAb on AAV pharmacology and potential harms in vivo. Therefore, we utilized multiple parameters involving the efficacy and safety of the tested drug to determine the exclusion criteria in future clinical trials, based on in vitro and in vivo tests.

Considering that the total neutralizing titer of polyclonal antibodies in serum is measured in practice, both polyclonal and monoclonal antibodies can be used for this purpose. We used a mouse anti-AAV9 monoclonal antibody (MoAb) in naïve mouse serum throughout the study to generate neutralization both in vitro and in vivo and also compared it with human purified immunoglobulin (Appendix A). The in vitro assay for NAb titer measuring was the same as for patient screening in future clinical trials, and the matrix effects of murine and human serum were comparable (Table 1). By transfusion with spiked sera, an animal model with anti-AAV9 NAb titers ranging from <1:100 to 1:1000 in vivo was obtained. According to our previous small-size study in Chinese PD patients (age < 10 years, *n* = 20), the anti-AAV9 NAb titers were all <1:1000, and the percentage of Nab-negative patients and titers below 1:100 was about 75% based on the in vitro assay used in this study (data not published). The animal model used in this study is a good simulation of the preexisting antibody state in PD patients.

Pompe disease is a neuromuscular disease due to a deficiency of the lysosomal enzyme GAA. Lack of the GAA enzyme causes glycogen accumulation in skeletal muscle, smooth muscle, and the heart. The effective transduction of the GAA gene to the heart and muscle is very important for drug efficacy. The liver is the organ with the most abundant AAV after systematic administration because of its rich blood supply; sustainably expressed GAA in the liver can be secreted into the blood and taken up by muscles and other peripheral organs to reinforce the efficacy [22,23]. As demonstrated in Figure 1, GC301 was widely distributed in the heart, liver, and muscles. The higher level of GAA biodistribution in the heart and liver than in muscles is consistent with our data on the mouse model for Pompe disease [9] and the findings reported by Wilson et al. [24]. Furthermore, we found different impacts of preexisting NAb on AAV transduction in the different tissues for the first time. A low level of NAb (titer <1:100) reduced the GAA expression in muscles, especially in the quadriceps, but had no or less impact in the heart and liver. Systemically delivered AAV vectors take a longer time to distribute to the muscles; as for whether the results in a greater impact of circulating preexisting antibodies or the presence of antigen–antibody complexes lead to increased immunogenicity in the muscles, resulting in more clearance, further investigation is needed.

Results from the current study showed that GC301 vector clearance was accelerated in peripheral blood, and transduction was completely blocked in the presence of a high level of preexisting NAb (a titer of about 1:1000 detected in vitro). The GAA activity only slightly increased, and no significant difference was found in mice with preexisting NAb at the titer of about 1:500 compared with NC. In the presence of moderate levels of preexisting NAb, at titers between 1:100 to 1:300 corresponding to the groups pretreated with MoAb at 3 and 7 μg/mL, gene copies and transduction of GC301 decreased in the heart and liver, especially in the liver, but were still significantly enhanced compared with NC group. Gregory D Hurlbut et al. used mice pretreated with saline or serially diluted inhibitory nonhuman primate (NHP) sera as a model to demonstrate that liver AAV8-αgal vector copy number fell off rapidly for titers above ~1:40 at low vector dose (2 × 10^12^ drp/kg), whereas for the higher dose (2 × 10^13^ drp/kg), this fall-off occurred for titers above ~1:640 [25]. In our study, we did not observe a dose-dependent difference at a low level of preexisting NAb. GC301 efficiently transduced the liver, heart, and muscles at both high and low dosages in mice after 1.5 μg/mL MoAb-pretreatment, at a titer of <1:100. We speculate that this is because the low dose (3 × 10^13^ vg/kg) used in this study is comparable to the higher doses in other studies.

The AAV-specific T cell responses tend to increase when the titer of preexisting NAb exceeds 1:200; whether it poses a safety risk needs to be determined further. Based on these results, we predict that low levels of NAb (<1:100) will not affect the efficacy and safety of GC301. Additionally, we also infused mice with human IVIG (Appendix A) and obtained a comparable preexisting NAb with the level of 1:47~1:68 (Appendix A) before 1.2 × 10^14^ vg/kg dose of GC301 injection, and found that little effect on the transduction efficiency in heart and liver as compared with those of non-pretreated mice on Day 28 (Appendix A). Given that enzyme activity does not need to reach normal levels to clear glycogen storage, a potential limitation of the current study is the use of Balb/C mice instead of a Pompe mouse model, which makes it impossible to evaluate if the lower transduction at higher antibody concentrations still enabled enough enzymes with therapeutic effects to be produced in the tissues. Considering that there is no long-term benefit from ERT in some Pompe disease patients, e.g., patients with high anti-GAA antibodies, and the potential immune responses against AAV9 could be controlled by immunosuppressive regimens which were widely used in gene therapy, the exclusion criteria of preexisting AAV9 NAb could be elevated to 1:200, even to 1:300. In fact, we have observed one IOPD patient with preexisting anti-AAV9 NAb (the titer was 1:168 at screening) achieve apparent clinical benefit after systematic delivery of GC301 at the dosage of 1.2 × 10^14^ vg/kg (paper in submit). In recent years, the important role of preexisting neutralizing antibodies in activating complement was observed [26,27,28]; the risk should be monitored and mitigated by using adequate immunosuppression strategies when dosing seropositive patients with vectors.

More and more clinical and preclinical data has indicated that low levels of preexisting AAV Ab could be overwhelmed by excessively high vector dosage and have minor effects on the therapeutic gene transduction [3,24,29]. A quantitative assessment of the role of preexisting NAbs in gene therapy efficacy and safety is required for the successful clinical development of AAV vectors. We combined the in vivo model with in vitro tests and applied multiple parameters to evaluate the impact of different levels of preexisting NAb, which could help us to determine the exclusion criteria in future clinical trials. However, caution needs to be placed on host factors that may impact outcomes.

## Figures and Tables

**Figure 1 viruses-16-00400-f001:**
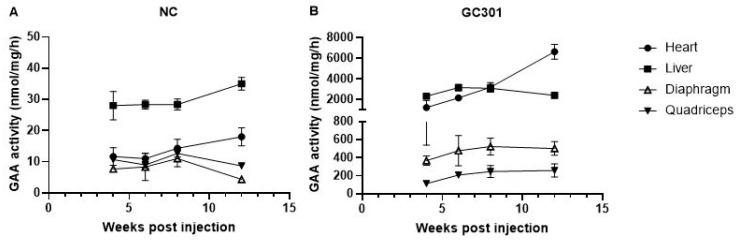
GAA activity in the heart, liver, and muscles of Balb/c mice post-rAAV9-coGAA injection. (**A**) Mice injected with 100 μL of PBS were used as negative control (NC), *n* = 5. (**B**) Mice were injected with rAAV9-coGAA (GC301) at the dose of 1.2 × 10^14^ vg/kg, *n* = 5.

**Figure 2 viruses-16-00400-f002:**
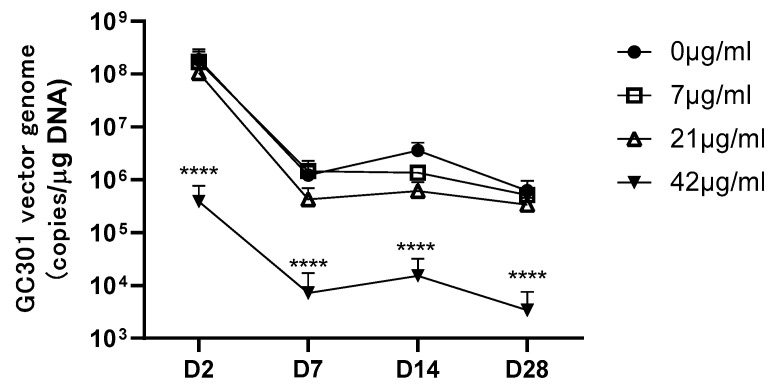
The vector genome copies in blood of anti-AAV9 MoAb-transferring mice post-GC301 injection at the dose of 1.2 × 10^14^ vg/kg. Mean and SD were displayed in the graph with 5~6 mice in each group. Statistical significance of vector genome among groups was determined using two-way ANOVA and multiple comparisons after log transform of original results. **** *p* < 0.0001 compared with anti-AAV9-MoAb negative group (0 μg/mL).

**Figure 3 viruses-16-00400-f003:**
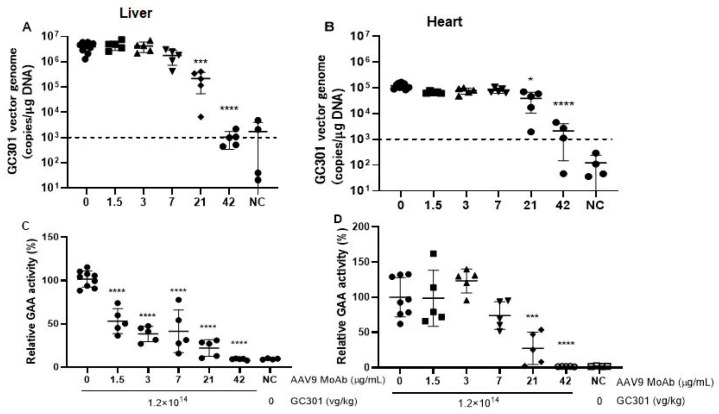
The vector genome copies and GAA activity in the liver and heart of anti-AAV9 MoAb-transferring mice 4 weeks post 1.2 × 10^14^ vg/kg rAAV9-coGAA injection. The vector genome copies in liver (**A**) and heart (**B**) were analyzed by ddPCR; the dash line is the limit of detection (LOD). The GAA enzyme activity was detected in tissue homogenate and was shown as relative GAA activity to the mean of AAV9-MoAb negative (0 μg/mL) mice, which were transferred with negative mouse serum and then received the same vector dose as that tested in liver (**C**) and heart (**D**). Statistical significance among the groups was determined using one-way ANOVA analysis and multiple comparisons based on the log transform on the data of vector genome copies and raw data of GAA activity. * *p* < 0.05, *** *p* < 0.001, **** *p* < 0.0001 compared with the results of anti-AAV9-MoAb negative group.

**Figure 4 viruses-16-00400-f004:**
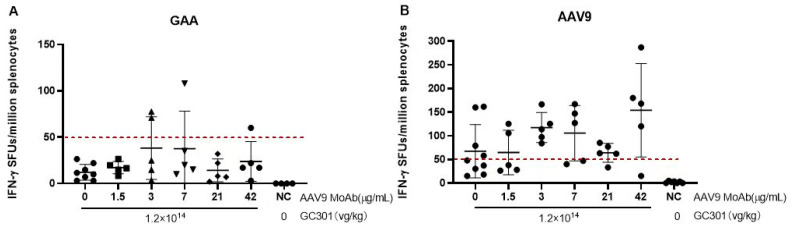
GAA (**A**) and AAV9 (**B**) specific T cell responses. The IFN-γ positive spots from peptides-stimulated splenic lymphocytes at 4 weeks after receiving GC301 at the dose of 1.2 × 10^14^ vg/kg by ELISPOT assay. Splenic lymphocytes were isolated using EZ-SepTM mouse lymphocyte separation medium. A total of 3 × 10^5^ cells were stimulated by the peptide pools, and less than 30 peptides per well with each peptide at the final concentration of 2.5 μg/mL. The spot-forming units (SFUs) were counted and calculated per one million cells.

**Figure 5 viruses-16-00400-f005:**
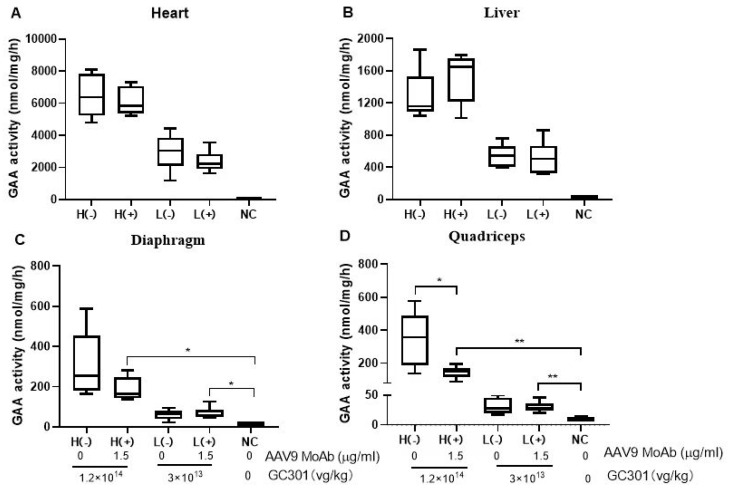
Efficient transductions of rAAV9-coGAA in 1.5 μg/mL MoAb-pretreated mice at both high and low dosages. The GAA activity of heart (**A**), liver (**B**), diaphragm (**C**), and quadriceps (**D**) were measured at 7 weeks after GC301 injection at the dosage of 1.2 × 10^14^ or 3 × 10^13^ vg/kg. The data are shown in Min to Max. * *p* < 0.05, ** *p* < 0.01 by unpaired Student’s *t* test.

**Table 1 viruses-16-00400-t001:** The titers of anti-AAV9 NAb in the sera of spiked humans, spiked mice, and passively immunized mice.

Spiked Anti-AAV9 MoAb (μg/mL)	AAV9 NAb Titer (Reciprocal Dilution, 1:x)		
	Spiked Human Sera	Spiked Mice Sera (Before Infusion)	Geometric Mean of Passively Immunized Mice Sera * (95% CI)
0	<1:20	<1:20	<1:20
1.5	1:431	1:456	1:60 (1:52~1:69)
3	1:729	1:997	1:193 (1:158~1:255)
7	1:2871	1:2976	1:290 (1:176~1:479)
21	1:8167	1:9823	1:481 (1:377~1:612)
42	1:20,884	1:26,671	1:851 (1:658~1:1101)

* The sera were collected 2 h after infusion with 100 μL spiked mice sera, *n* = 5~6 per each group. MoAb, monoclonal antibody. The experiments shown here were repeated and analyzed twice.

## Data Availability

Materials and protocols will be distributed to qualified scientific researchers for noncommercial, academic purposes. The rAAV9-coGAA vector (GC301) and the vector sequence are part of an ongoing development program, and they will not be shared.

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
