# Peer review of "Using an In Vivo Mouse Model to Determine the Exclusion Criteria of Preexisting Anti-AAV9 Neutralizing Antibody Titer of Pompe Disease Patients in Clinical Trials"

_viruses, 2024, doi:10.3390/v16030400_

Round 1
Reviewer 1 Report
Comments and Suggestions for Authors
Author Response
Thanks very much for your comments and suggestions. We carefully revised our manuscript and formatted it according to the requirement of the research article. We thank your interest in our study and would like to give the following point-by-point response.
Comment 1:
Wang et al. present a study analysing the in vivo affect of anti-AAV9 neutralizing antibody on the efficacy of AAV9-GAA treatment for Pompe disease. The manuscript is generally well written and presented and the study is of importance to the scientific community, not only for the treatment of Pompe, but for consideration in other AAV clinical trials.
My one major concern is on the use of Balb/C mice instead of a Pompe mouse model. Given that enzyme activity does not need to reach normal levels to clear storage, I think it would have been beneficial to see if the lower transduction at higher antibody concentrations still enabled enough enzyme to be produced in the tissues. I think this needs to be discussed as a potential limitation of the current study.
Answer:
Your comments and suggestions are very appreciated. We discussed the point in the revised manuscript (Lines 359-363 in Page 9).
Comment 2:
The rest of my comments are minor for clarity or ease of reading.
Page 2, line 45, write out SMA in full (abbreviation has not been previously defined). Page 2, page 70, drivined?? driven
Page 2, line 74, please include the reference for “reported previously.”
Page 2, animal studies. I think a timeline would be useful here to make it clear to the reader the timepoints of the study.
Page 3, line 103, luciferase unit (not unite).
Page 3, line 105: please replace “certain” with the actual concentrations used.
Page 3, GAA enzyme activity. Was a control enzyme measured to show that differences observed were specific to GAA activity and not sample integrity?
Page 4, Figure 1. Please include units and number of mice per group in the Figure legend. Page 5, Figure 2. Y-axis title is illegible.
Page 6, Section 3.4. At what timepoint was this data from?
Page 6, Figure 3. Y-axes titles for A and B illegible. The µ symbol is also difficult to distinguish in the x- axis.
Page 7, line 267, this is the first mention of H group (and L group below this). This would be better defined in the methods and may be easier to understand if referred to as high dose and low dose groups. This can then be used to throughout the results instead.
Answer:
Thank you for the careful review. We have corrected the words or sentences as the reviewer pointed out.
Comment 3:
Minor issues throughout:
- There are a few changes in tense throughout. Eg, page 1, line 21 present tense is used
instead of past tense. Should be “tended,” “exceeded” and “had.”
- The use of subscript and superscript is missing in sections throughout.
- The axis titles for many of the figures are difficult to read. Seems to be some bolding?
Answer: Thank you for the careful review. We have corrected the tense and formatting in the revised manuscript, as the reviewer pointed out.
We are grateful to you and the reviewers for all suggestions in improving our manuscript. We checked our revised manuscript thoroughly, including the text, figures, and references. We hope the revised manuscript can be accepted for publication. Please do not hesitate to contact us for further clarifications.

Reviewer 2 Report
Comments and Suggestions for Authors
In this report, Wang et al. studied the impact of anti-AAV9 neutralizing antibody (Nab) titer on transduction of GC301 (an AAV9-based clinical gene therapy candidate for Pompe disease) in mice, and concluded that low levels of NAb titers (~1:100) had no effect on transduction in the heart and liver, and slightly decreased transduction in muscle.
A mouse anti-AAV9 monoclonal antibody (MoAb) was used throughout the study to generate neutralization both in vitro and in vivo. Use of human IVIg would better represent pre-existing Nabs, which has been a common method to study the impact of anti-AAV Nabs on in vivo transduction in mice. It seems authors’ rationale was “To avoid potential immunogenicity or interference of serum from monkeys or humans, …” (lines 306-307) However, such a hypothetical caveat in the context of studying anti-AAV Nab was not proven or studied in this manuscript. To the bare minimum, the experiments relating to Table 1 and Figure 3 should be repeated using human IVIg, and compare against the results obtained using MoAb.
Authors tried to cite literature to support their findings on low levels of pre-existing Nabs are permissive to AAV transduction (refs 3 and 4 in line 35, and ref 6 in line 338), but those publications do not contain similar results. Specifically:
- Ref 3 (Xue et al. Lancet Haematol 2022): In this study, Nabs in patients were 1:1 or 1:2, much lower than 1:100 in this manuscript. Authors vaguely cited it as a case of “low levels of pre-existing Nabs” (lines 34-35)
- Ref 4 (Von Drygalski et al. Blood Adv 2019): This study was about AAV5, which has a unique immunogenicity profile that remains to be further studied. Authors wrongly cited it as a relevant case for AAV8 (lines 34-35).
- Ref 6 (Fitzpatrick et al. Mol Ther Methods Clin Dev 2018): This report studied differences between AAV8 NAbs and binding antibodies (BAbs); what did not negatively impact on transduction was BAbs, whereas NAbs at 1:10 blocked transduction. Authors wrongly cited it as “similar results” pertaining to NAbs (lines 335-337)
Comments on the Quality of English LanguageQuality of English is generally acceptable.
Author Response
Thanks very much for the reviewer’s comments and suggestions. We carefully revised our manuscript and formatted it according to the requirement of the research article. We thank your interest in our study and would like to give the following point-by-point response.
Comment 1:
In this report, Wang et al. studied the impact of anti-AAV9 neutralizing antibody (Nab) titer on transduction of GC301 (an AAV9-based clinical gene therapy candidate for Pompe disease) in mice, and concluded that low levels of NAb titers (~1:100) had no effect on transduction in the heart and liver, and slightly decreased transduction in muscle.
A mouse anti-AAV9 monoclonal antibody (MoAb) was used throughout the study to generate neutralization both in vitro and in vivo. Use of human IVIg would better represent pre-existing Nabs, which has been a common method to study the impact of anti-AAV Nabs on in vivo transduction in mice. It seems authors’ rationale was “To avoid potential immunogenicity or interference of serum from monkeys or humans, …” (lines 306-307) However, such a hypothetical caveat in the context of studying anti-AAV Nab was not proven or studied in this manuscript. To the bare minimum, the experiments relating to Table 1 and Figure 3 should be repeated using human IVIg, and compare against the results obtained using MoAb.
Answer:
Thank you very much for pointing out our unclear statements on experiment-design. We have changed to the following statement:
Considering that the total neutralizing titer of polyclonal antibodies in serum is measured in practice, both polyclonal and monoclonal antibodies can be used for this purpose. Unlike other studies that use IVIG or AAV immunized animal serum for modeling, we used a mouse anti-AAV9 monoclonal antibody (MoAb) in naïve mice serum throughout the study to generate neutralization both in vitro and in vivo. (Lines 309-313 in Page8-9.)
Comment 2:
Authors tried to cite literature to support their findings on low levels of pre-existing Nabs are permissive to AAV transduction (refs 3 and 4 in line 35, and ref 6 in line 338), but those publications do not contain similar results. Specifically:
- Ref 3 (Xue et al. Lancet Haematol 2022): In this study, Nabs in patients were 1:1 or 1:2, much lower than 1:100 in this manuscript. Authors vaguely cited it as a case of “low levels of pre-existing Nabs” (lines 34-35)
- Ref 4 (Von Drygalski et al. Blood Adv 2019): This study was about AAV5, which has a unique immunogenicity profile that remains to be further studied. Authors wrongly cited it as a relevant case for AAV8 (lines 34-35).
- Ref 6 (Fitzpatrick et al. Mol Ther Methods Clin Dev 2018): This report studied differences between AAV8 NAbs and binding antibodies (BAbs); what did not negatively impact on transduction was BAbs, whereas NAbs at 1:10 blocked transduction. Authors wrongly cited it as “similar results” pertaining to NAbs (lines 335-337)
Answer: Thank you for your very detailed review. We have deleted ref 3 and ref 6, and corrected the information of ref 4 (Line 34 in Page 1,Ref 3 after revision).
We are grateful to you and the reviewers for all suggestions in improving our manuscript. We checked our revised manuscript thoroughly, including the text, figures, and references. We hope the revised manuscript can be accepted for publication. Please do not hesitate to contact us for further clarifications.

Reviewer 3 Report
Comments and Suggestions for Authors
Wang et al. investigated the effect of pre-existing anti-AAV neutralizing antibodies on the efficacy of AAV-mediated gene therapy. Specifically, different doses of mouse anti-AAV9 specific monoclonal antibodies were injected into groups of mice prior to AAV-vector injection, and vector genomes and transgene expression in liver, heart, and muscle was quantified at different times after vector delivery. They show that medium to high levels of neutralizing antibodies can efficiently block AAV transduction. Low levels of neutralizing antibodies had only a minor effect on transduction efficiency.
While novelty is limited, the work is solid and the data important. One main critique is that the experiments have been performed only a single time, at least according to the information given in the manuscript. Minor issue: Some of the labels in the graph require reformatting.
Comments on the Quality of English Language
The text should be edited for language.
Author Response
Thanks very much for the reviewer’s comments and suggestions. We carefully revised our manuscript and formatted it according to the requirement of the research article. We thank your interest in our study and would like to give the following point-by-point response.
Wang et al. investigated the effect of pre-existing anti-AAV neutralizing antibodies on the efficacy of AAV-mediated gene therapy. Specifically, different doses of mouse anti-AAV9 specific monoclonal antibodies were injected into groups of mice prior to AAV-vector injection, and vector genomes and transgene expression in liver, heart, and muscle was quantified at different times after vector delivery. They show that medium to high levels of neutralizing antibodies can efficiently block AAV transduction. Low levels of neutralizing antibodies had only a minor effect on transduction efficiency.
While novelty is limited, the work is solid and the data important. One main critique is that the experiments have been performed only a single time, at least according to the information given in the manuscript. Minor issue: Some of the labels in the graph require reformatting.
Answer:
Thank you for the careful review and comments. In fact, the experiment shown in Table 1 was repeated and analyzed twice, and the information has been added to the table (Line 199). Other animal experiments were partially repeated. The study after GC301 administration consisted of 3 independent animal experiments, the results of Experiment 1 are shown in Figure 1, the results of Experiment 2 are shown in Figures 2 to 4, and the results of Experiment 3 are shown in Figure 5. The high-dose group (1.2E+14 vg/ml) without pre-existing NAb (which was shown as H(-) in Figure 5), and negative control (NC) group were included in each experiment. The high-dose group with a low level of pre-existing NAb (pretreated with 100 μl of 1.5 μg/ml AAV9 MoAb, which was shown as H(+) in Figure 5) was also repeated in Experiment 2 and 3, and the results were displayed in Figure 3 and Figure 5 respectively.
The labels in the graph have been reformatted.
We are grateful to you and the reviewers for all suggestions in improving our manuscript. We checked our revised manuscript thoroughly, including the text, figures, and references. We hope the revised manuscript can be accepted for publication. Please do not hesitate to contact us for further clarifications.

Round 2
Reviewer 2 Report
Comments and Suggestions for Authors
Authors did not adequately address comment 1 (use of human IVIg instead of a mouse anti-AAV9 monoclonal antibody), but downplayed the issue as "unclear statements".
As to comment 2 (3 literature citing issues), I appreciate authors' revision by deleting 2 references and corrected information for the 3rd.
Comments on the Quality of English LanguageNo major English language issue was identified.
Author Response
Dear reviewer,
Re: Using in vivo mouse model to determine the exclusion criteria of preexisting anti-AAV9 neutralizing antibody titer of Pompe disease patients in clinical trial
Thanks for the reviewers' comments and suggestions. We carefully revised our manuscript. As to the suggestion from the reviewer as mentioned in the following comment, we attached the experimental data on Balb/C mice infused with a comparable neutralizing amount of IVIG as 1:100 anti-AAV9 MoAb tested in our study and compared their effects.
Comment: There are legitimate concerns that this study has been performed with only a monoclonal antibody and not IVIG which is more representative of a natural humoral anti-AAV immune responses. The authors did not address this point sufficiently in this revision and they should address this concern experimentally. Details of the monoclonal antibody should also be provided.
Answer:
We followed your suggestions and supplemented the experimental data using a commercial IVIG at a stocking concentration, 75mg/mL. We also discussed the data in the revised manuscript (Page9, Lines 314 to 316 and 362 to 365).
The anti-AAV9 MoAb tested in our study was generated by ourselves using hybridoma technique from empty AAV9 particle-immunized Balb/C mice. The MoAb is specifically neutralized AAV9 serotype with an IC50 of 20 ng/ml in vitro. We added the information in the revised manuscript (Page2, Lines 88 to 90)
We are grateful to you and the reviewers for all suggestions in improving our manuscript. We checked our revised manuscript thoroughly, including the text, figures, references, and supplementary information. We hope the revised manuscript can be accepted for publication. Please do not hesitate to contact us for further clarifications.

Round 3
Reviewer 2 Report
Comments and Suggestions for Authors
I appreciate the additional IVIG experiment conducted by authors to address my final concern.
Comments on the Quality of English LanguageMay consider editing for minor language issues.